# Safety and Efficacy of Carbon-Ion Radiotherapy for Elderly Patients with High-Risk Prostate Cancer

**DOI:** 10.3390/cancers14164015

**Published:** 2022-08-19

**Authors:** Yuichi Hiroshima, Hitoshi Ishikawa, Yuma Iwai, Masaru Wakatsuki, Takanobu Utsumi, Hiroyoshi Suzuki, Koichiro Akakura, Masaoki Harada, Hideyuki Sakurai, Tomohiko Ichikawa, Hiroshi Tsuji

**Affiliations:** 1QST Hospital, National Institutes for Quantum Science and Technology, Chiba 263-8555, Japan; 2Department of Radiation Oncology, Proton Medical Research Center, University of Tsukuba Hospital, Tsukuba 305-8576, Japan; 3Department of Radiology, Chiba University Graduate School of Medicine, Chiba 260-8670, Japan; 4Department of Urology, Toho University Sakura Medical Center, Chiba 285-8741, Japan; 5Department of Urology, Japan Community Health-Care Organization Tokyo Shinjuku Medical Center, Tokyo 162-8543, Japan; 6Department of Urology, Chiba University Graduate School of Medicine, Chiba 260-8670, Japan

**Keywords:** carbon-ion radiotherapy, elderly patients, high-risk prostate cancer, radiotherapy, particle beam therapy

## Abstract

**Simple Summary:**

As the population ages, the number of elderly prostate cancer patients is increasing, and the choice of treatment options for elderly patients with poor general condition is becoming more difficult. The aim of our retrospective study is to assess the clinical outcomes after carbon-ion radiotherapy for elderly patients with high-risk prostate cancer. We compared 173 patients ≥75 years as the elderly group and 754 patients <75 years as the young group. Disease-specific and biochemical relapse-free survivals did not differ significantly between the young and elderly groups, and there were also no significant differences in adverse events between the two groups. Although this study is retrospective, carbon-ion radiotherapy may be a safe and effective treatment for elderly high-risk prostate cancer patients.

**Abstract:**

Carbon-ion radiotherapy (CIRT) is a high-dose intensive treatment, whose safety and efficacy have been proven for prostate cancer. This study aims to evaluate the outcomes of CIRT in elderly patients with prostate cancer. Patients aged 75 years or above at the initiation of CIRT were designated as the elderly group, and younger than 75 years as the young group. The overall survival (OS), disease-specific survival (DSS), biochemical control rate (BCR), biochemical relapse-free survival (BRFS), and adverse events were compared between the elderly and young patients with high-risk prostate cancer treated with CIRT. The elderly group comprised 173 of 927 patients treated for high-risk prostate cancer between April 2000 and May 2018. The overall median age was 69 (range: 45–92) years. The median follow-up period was 91.9 (range: 12.6–232.3) months. The 10-year OS, DSS, BCR, and BRFS rates in the young and elderly groups were 86.9%/71.5%, 96.6%/96.8%, 76.8%/88.1%, and 68.6%/64.3%, respectively. The OS (*p* < 0.001) was longer in the younger group and the BCR was better in the elderly group (*p* = 0.008). The DSS and BRFS did not differ significantly between the two groups. The rates of adverse events between the two groups did not differ significantly and no patient had an adverse event of Grade 4 or higher during the study period. CIRT may be as effective and safe in elderly patients as the treatment for high-risk prostate cancer.

## 1. Introduction

In Japan, the incidence of prostate cancer has increased in recent years, partially due to the popularity of prostate-specific antigen (PSA) screening; the reported incidence was 92,021 in 2018, making it the most commonly diagnosed cancer, surpassing stomach, colorectal, and lung cancers [1]. The incidence rate increased with age, with 23.3 cases per 100,000 persons in the 50–54 age years group and 75.1 cases in the 55–59 age years group, with a particularly sharp increase in those aged 60 and over, and a high rate of about 650 cases in those aged 75 and over [1]. Japan is one of the fastest aging countries in the world, and it is clear that the incidence of prostate cancer will rise with the increase in the number of elderly individuals. On the other hand, the decline in the performance status (PS) and activities of daily living (ADLs) in elderly individuals prior to treatment often makes healthcare providers, patients, and their families hesitant to consider curative treatment even after the diagnosis of malignancy [2,3,4].

There are two types of treatment for prostate cancer: curative and noncurative. For the elderly, androgen deprivation therapy (ADT) may be chosen as noncurative treatment based on the predicted prognosis. However, with the increase in life expectancy, prostate cancer may become resistant to castration and, progress and metastasize before death from other diseases. In particular, prostate cancer is known to metastasize to bones, and bone fractures in the elderly people have been reported to reduce prognosis as well as quality of life [5,6]. Against this background, minimally invasive curative treatment that can be undergone by the elderly is desired.

The two principal types of curative treatment for localized prostate cancer are surgery and radiotherapy. The use of robot-assisted laparoscopic prostatectomy (RALP) has become more widespread, while the precision of radiotherapy, especially external-beam radiotherapy (EBRT), has increased with the development of image-guided radiotherapy [7,8,9]. Surgery is usually not performed for men over 75 years of age due to the high risk of perioperative mortality, whereas radiotherpay is minimally invasive and can be performed for those over 75 years of age if they are in a good general condition [10]. Therefore, the patient population undergoing radiotherapy tends to be relatively older compared to that undergoing surgery, and there is potential for further growth in the number of patients, owing to the aging of the population. However, radiotherapy in the elderly is characterized by a greater likelihood of deterioration in the quality of life affecting the performance of daily activities, especially during long treatment periods, which is exacerbated by adverse events, such as frequent urination during treatment and hematuria and hematochezia after treatment [11].

Particle beam therapy, an EBRT technique, uses atoms accelerated at a high speed by an accelerator. The efficacy of particle beam therapy has also been reported for the treatiment of pancreatic, liver, and other cancers [12,13,14,15,16,17,18,19]. Particle beam therapy can be further categorized into proton beam therapy and carbon-ion radiotherapy (CIRT). CIRT is known to have a smaller peripheral effect and greater biological effectiveness compared to proton beam therapy. Therefore, a previous study reported that CIRT can facilitate a further reduction in the irradiation to the surrounding organs, including the rectum, thereby reducing adverse events and enabling the effective treatment of prostate cancer over a shorter therapeutic period [20,21].

CIRT has been used for the treatment of prostate cancer since 1995 [20]. Although the previous studies have reported on the treatment results and toxicity, none have focused on elderly patients [22,23,24,25,26,27,28,29,30,31,32,33,34]. Therefore, this study aims to assess and report the outcomes and toxicity of CIRT for prostate cancer in patients aged 75 years and older, compared to those in their younger counterparts.

## 2. Materials and Methods

### 2.1. Patients

We retrospectively searched for patients who underwent CIRT for prostate cancer at our institution between April 2000 and May 2018. The eligibility criteria for CIRT at our institution have been described in previous studies [28,29,32,33]. The inclusion criteria for this study were as follows: (1) prostate cancer diagnosis confirmed by biopsy, whose Gleason score (GS) was evaluated by a pathologist; (2) patients who met the diagnostic criteria for high-risk disease based on the National Comprehensive Cancer Network risk classification: initial PSA ≥ 20 ng/mL, or T3a-T3bN0M0, or GS ≥ 8 [35]; (3) endocrine therapy started 3–6 months before the initiation of CIRT; (4) patients aged 20 years or older at the start of therapy; and (5) PS in the range of 0–2. The exclusion criteria for this study included the following: (1) lymph node or distant metastasis prior to the initiation of treatment, (2) previous treatment of prostate cancer with a modality other than ADT, (3) active and refractory infections of the prostate, and (4) presence of other active cancers. The participants provided informed consent for participation or had the opportunity to opt-out of the study.

### 2.2. ADT

ADT was based on the luteinizing hormone-releasing hormone (LHRH) analog or castration combined with anti-androgen therapy. The LHRH analog alone was also acceptable in case of liver dysfunction. ADT was to be started at least 3 months prior to the start of CIRT and continued for the duration of treatment and for at least 1 year after completion. However, in cases where it was difficult to continue the treatment due to side effects or other reasons, discontinuation of the treatment was allowed.

### 2.3. CIRT

The details of the CIRT regimen for prostate cancer have been reported previously [20,23]. In the past, thermoplastics were used to fix the head, trunk, and pelvis, but due to advances in image guidance equipment, treatment is gradually being performed using only a pelvic fixture. CT is obtained in 2 mm slices and combined with MRI to improve prostate positioning accuracy. In addition, about 100 mL of urine was stored in the bladder at the time of CT imaging, and ultrasound was used to confirm that the same amount of urine was stored at the time of treatment. Laxatives were also occasionally used to keep the rectum as empty as possible. Briefly, the entire prostate and proximal third or half of the seminal vesicle (SV) for T1–T3a and as much of the SV as possible for T3b were designated as the clinical target volume. The radiation dose of CIRT was expressed as the photon equivalent dose (Gy) (i.e., the relative biological effect (RBE)-weighted absorbed dose) and defined as the physical dose multiplied by the carbon-ion RBE [36]. Total dose and number of fractions (fr) were treated at 66 Gy/20 fr from January 1998 to August 2005, 63 Gy/20 fr from September 2005 to August 2007, 57.6 Gy/16 fr from September 2007 to March 2013, 54 Gy/16 fr from September 2011 to July 2012, and 51.6 Gy/12 fr from April 2013 to the present, respectively, as the clinical trials were conducted at different times when treatment was performed [20]. In April 2013, CIRT using the scanning irradiation method was started instead of the passive irradiation method [23]. Treatment planning for CIRT was performed with HIPLAN (NIRS, Chiba, Japan) until March 2013, and after that, XiO-N (ELEKTA, Stockholm, Sweden). Patient background and treatment details and their respective numbers and percentages are shown in Table 1.

### 2.4. Evaluation

After the completion of CIRT, patients were required to undergo a follow-up examination at 3-month intervals for 2 years initally, and at 6-month intervals thereafter. Blood samples were collected during each visit. After the completion of CIRT, patients were required to undergo a follow-up examination at 3–6-month intervals for the first 5 years, and at 6–12-month intervals thereafter if no sign of the recurrences were observed. Even after a 10-year follow-up examination, patients basically came to the hospital as long as they could in order to evaluate the risk of subsequent primary second cancers in patients with prostate cancer after CIRT [37], but patients whose regular follow-up was interrupted were contacted directly by mail or telephone to check on their condition. Blood samples were collected during each visit. Post-treatment biochemical recurrence (BR) was defined as a PSA value exceeding 2.0 ng/mL over the minimum value during treatment according to the Phoenix definition [35]. In the event of BR, bone scintigraphy, computed tomography (CT), and magnetic resonance imaging (MRI) were performed to confirm the presence of recurrence. Local recurrence was defined as regrowth, as observed on CT and MRI. Adverse events were assessed in accordance with the Common Terminology Criteria for Adverse Events (CTCAE) version 4.0, and the previous studies [38,39,40]. Acute-phase adverse events were defined as those occurring within 3 months of treatment; and late-phase adverse events were defined as those occurring 3 months after treatment.

### 2.5. Statistical Analysis

The Mann–Whitney U and χ^2^ tests were used for the statistical analysis of the continuous and categorical variables, respectively. The Kaplan–Meier method was used to analyze overall survival (OS), disease-specific survival (DSS), biochemical control rate (BCR), and biochemical relapse-free survival (BRFS), and the log-rank test was used for a comparison of these parameters. SPSS software version 28.0 (IBM Inc., Armonk, NY, USA) was used for all analyses. *p*-values < 0.05 were considered statistically significant.

### 2.6. Ethics Approval

This research was conducted in accordance with the Declaration of Helsinki (1964) and the subsequent Code of Ethics, and the study design was approved by the Institutional Review Board (permit number: N21-005).

## 3. Results

### 3.1. Patient Characteristics

A total of 927 patients with prostate cancer who started CIRT between April 2000 and May 2018 met the eligibility criteria. Of these, 173 were aged 75 years or older and placed in the elderly group, and others who were younger than 75 years were defined as the young group. The median age of the entire patient population was 69 years (range: 45–92 years), and the median PSA level before treatment initiation was 16.7 ng/mL (range: 2.1–329.7). The other patient characteristics are presented in Table 1. There were no significant differences in the patient factors between the young and elderly groups.

The median follow-up duration for the entire patient population and elderly group was 91.9 (range: 12.6–232.3 months) and 84.4 (range: 13.5–214.8 months) months, respectively. During the observation period, there were 147 and 38 instances of all-cause motality in the entire patient population and elderly group, respectively, including 31 and 4 deaths due to prostate cancer, respectively. Recurrences during the observation period were as follows: BCR was observed in 161 patients (17.4%) in the entire patient population and 15 patients (8.7%) in the elderly group. The distributions of local, regional, and distant metastases are shown in Figure 1. One patient developed regional recurrence and another developed regional and distant recurrences concurrently, neither of whom had BCR.

### 3.2. Adverse Events

The gastrointestinal (GI) and genitourinary (GU) adverse events that occured during the observation period are shown in Table 2. CTCAE grades 4 or 5 adverse events were not observed in any patient during the study period. The occurence of adverse events, especially the higher grades, in the elderly group was similar or lower than that in the young group.

### 3.3. Outcomes

The OS, DSS, BCR, and BRFS in the young group at 5 and 10 years were 96.3% and 86.9%, 99.0% and 96.6%, 89.7% and 76.8%, and 87.1% and 68.6%, respectively (Figure 2). The corresponding rates in the elderly group were 92.6% and 71.5%, 98.2% and 96.8%, 93.4% and 88.1%, and 87.4% and 64.3%, respectively.

The OS was significantly longer in the younger group, while the BCR was significantly better in the elderly group (*p* < 0.001 and *p* = 0.008, respectively). The DSS and BRFS did not significantly differ between the two groups (*p* = 0.736 and *p* = 0.597, respectively).

The survival times according to age in the present study were compared with the expected life expectancy by age for Japanese men in 2020 [41]. No obvious differences were observed in the two groups under 74 years of age. However, the survival time was better for those aged 75–79 and 83 years in the group treated with CIRT (Figure 3).

## 4. Discussion

This study examined the efficacy and toxicity of CIRT for high-risk prostate cancer in patients aged above and below 75 years. The OS was significantly better in the young group, while the DSS and BRFS did not differ significantly between the two groups. This suggests that while the curative effect of CIRT for prostate cancer was equivalent for both groups, the OS was shortened by other diseases that occur more frequently in the elderly population, such as death due to other cancers and cardiovascular events. On the other hand, the BCR was significantly better in the elderly, although there were no differences in the prostate cancer risk category and treatment methods between the two groups. The difference in BCR may be attributed to the age-related decline in androgen secretion. Androgens are among the factors that promote the progression of prostate cancer, and their secretion is also known to decrease with age [42,43,44]. Thus, there may be background factors that may decrease the likelihood of prostate cancer recurrences in the elderly, or it may take a long time for the PSA to rise sufficiently to the diagnostic range.

More hypofractionated radiotherapy for prostate cancer is becoming more common with intensity-modulated radiotherapy and stereotactic body radiotherapy. A meta-analysis, including a phase III prospective study, found 14.6% of GI adverse events and 19.4% of GU adverse events to be grade 2 or higher, which is a higher percentage than either group in this study [45]. Another meta-analysis of prospective studies found 1.1% of GI adverse events and 2.0% of GU adverse events to be grade 3 or higher, also a higher rate than in the present study. Thus, CIRT may be safer than existing EBRT [46]. No grade 4 or higher adverse events were observed in this study, and the occurence of adverse events in each category was either similar or, in some cases, slightly higher in young group. Although the OS of the elderly group was shorter than that of the young group, the duration of follow-up was comparable. These results suggest that the frequency and severity of adverse events may be similar between the elderly and young groups. The occurence of cardiovascular disease, such as atrial fibrillation, is often higher in elderly patients, who are frequently prescibed anticoagulants, exposing to a higher risk for bleeding-related adverse events [47,48,49]. These medications were not withdrawn or discontinued at the time of CIRT, and patients were allowed to continue taking them afterwards. The fact that there was no apparent difference in the frequency of adverse events, especially the GU and GI adverse events, suggests that CIRT is a safe treatment modality for elderly as well as young patients.

A comparison of the mean OS rates and expected life expectancy of the patient groups included in the study with the general population revealed that OS for patients treated with CIRT was similar or superior, especially in the 75–79-year-old group. This suggests that even elderly patients with high-risk prostate cancer can survive up to their respective age-adjusted mean life expectancies after radical treatment with CIRT. The encouraging results in the 76–80-year-old age group in the present study may be partially attributed to the fact that, historically, age was one of the criteria for surgery in Japan. Urologists were first consulted when the PSA screening results required further investigation; moreover, urologists often prescribed surgery for patients up to 75 years of age, when the expected life expectancy was 10 years or less, before the widespread use of RALP [7]. Therefore, patients aged older than 76 years who underwent CIRT were referred to our hospital because they were not considered as suitable surgical candidates due to their advanced age. However, it is possible that the results of the present study are the product of enrolling patients whose PS and ADLs are better than those of their (age-matched) contemporaries. At present, RALP is widely used, and is performed even for patients aged 76–80 years if their PS and ADLs are good. The accumulation of patients in the future will clarify whether this is the reason for good life expectancy in the 76–80-year-old age group observed in the study.

The elderly population continues to grow in Japan, with a corresponding rise in the incidence of prostate cancer. In the real-world clinical setting, younger patients may be ready for radical treatment, while elderly patients are often hesitant because of concerns about worsening of the general condition and adverse events after treatment, which may discourage healthcare providers, the patient, and family members from proceeding with treatment. In this context, guidelines and reports have been devised for elderly cancer patients [2,4,11]. The common desires binding all patients is the wish to complete the treatment in as short a time as possible with high precision, and in as few days as possible, especially for elderly patients. In this regard, CIRT is considered to be a guideline-compliant treatment, which offers highly precise physical dose distribution and allows for hypofractionation. Further advancements in hypofractionation in the future may enable even more elderly patient-friendly treatment.

The limitations of this study included its retrospective design, bias arising from the patients’ background, relatively small sample size of the elderly group (n = 173), single-center setting, variation in the time of CIRT initiation, and differences in the total dose and fractionation. On the other hand, no other study has examined the prognosis after CIRT by exclusively focusing on elderly patients with prostate cancer. Furthermore, the GS, which varied depending on each individual pathologist, was evaluated by only one pathologist at our institution for more than 25 years, rendering the reliability of risk classification to be high. In addition, this study included missing data from 83 (8.9%) of the 927 patients (43 in the young group and 40 in the elderly group) because of being lost to follow-up until 10 years after CIRT. However, the median follow-up times for the 43 young and 40 elderly patients were 81.3 and 85.3 months, respectively. Since these follow-up times were similar to those of the entire cohort in the present study, the impact of the missing data on comparing treatment outcomes of the young and elderly groups may be limited. Further studies with a longer follow-up period, larger sample size, and prospective design are needed to validate the results of this study.

## 5. Conclusions

The evaluation of the efficacy and toxicity of CIRT in elderly patients with prostate cancer revealed that CIRT may yield a good therapeutic effect on these patients, comparable to that of their younger counterparts.

## Figures and Tables

**Figure 1 cancers-14-04015-f001:**
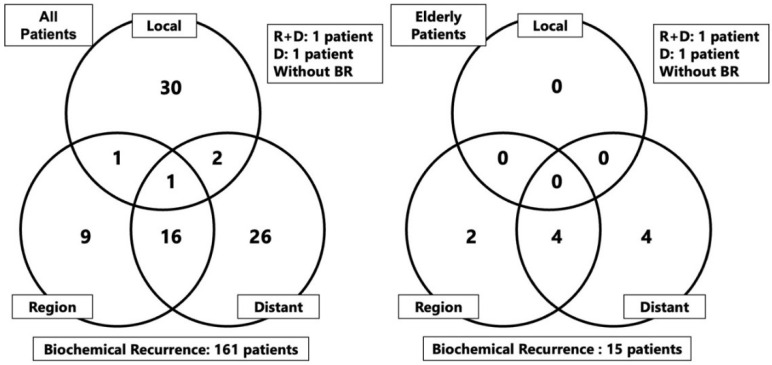
Form of recurrences after carbon-ion radiotherapy in the overall study population and elderly group. Abbreviation: R: region, D; distant, BR: biochemical recurrence.

**Figure 2 cancers-14-04015-f002:**
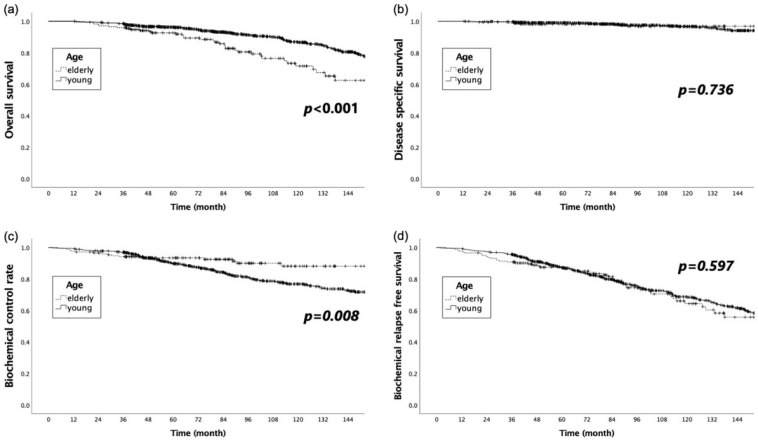
Comparison of the Kaplan–Meier curves for (**a**) overall survival, (**b**) disease-specific survival, (**c**) biochemical recurrence control rate, and (**d**) biochemical relapse-free survival for the young and elderly groups.

**Figure 3 cancers-14-04015-f003:**
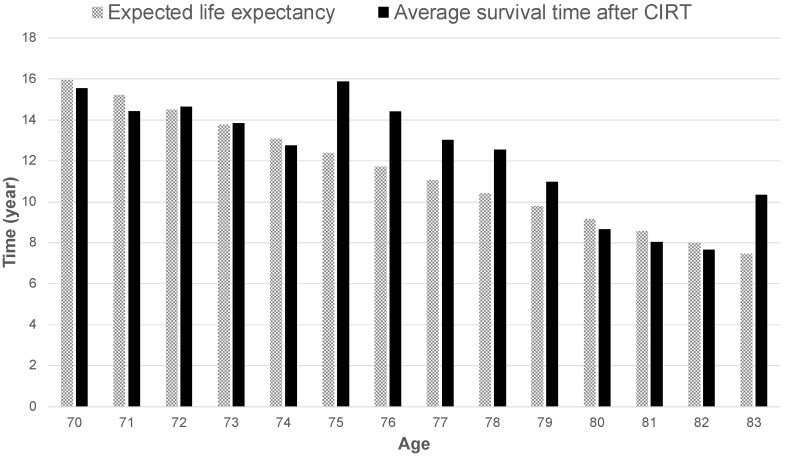
Expected life expectancy by age in Japan in 2020 compared to the average overall survival time by age at the start of carbon-ion radiotherapy.

**Table 1 cancers-14-04015-t001:** Patient, tumor, and treatment characteristics.

	Overall Study Population	Young Group	Elderly Group	*p*-Value
age	Median: 69 (range 45–92)	Median: 67 (range: 45–74)	Median: 77 (range: 75–92)	
Initial PSA (ng/mL)	Median: 16.7 (range 2.1–329.7)	Median: 16.92 (range: 2.1–320.0)	Median: 16.0 (range: 3.3–329.7)	0.590
Clinical T stage 1c/2a/2b/2c/3	139/159/22/149/458 15.0%/17.2%/2.4%/16.1%/49.4%	112/122/20/115/385 14.9%/16.2%/2.7%/15.3%/51.1%	27/37/2/34/73 15.6%/21.4%/1.2%/19.7%/42.2%	0.110
GS sum 5/6/7/8/9/10	2/45/301/212/364/3 0.2%/4.9%/32.5%/22.9%/39.3%/0.3%	1/37/258/163/292/3 0.1%/4.9%/34.2%/21.6%/38.7%/0.4%	1/8/43/49/72/0 5.8%/4.6%/24.9%/28.3%/41.6%/0%	0.114
Primary GS ~3/4/5	255/621/51 27.5%/67.0%/5.5%	211/502/41 28.0%/66.6%/5.4%	44/119/10 25.4%/68.8%/5.8%	0.792
Total ADT duration (months)	Median: 20.9 (range: 0.6–63.6)	Median: 22.0 (range: 0.6–63.5)	Median: 18.2 (range: 0.6–63.6)	0.265
Total dose (Gy)/fraction 51.6/20, 54/16, 57.6/16, 63/20, 66/20	311, 4, 331, 128, 153 33.5%, 0.4%, 35.7%, 13.8%, 16.5%	251, 3, 267, 104, 129 33.3%, 0.4%, 35.4%, 13.8%, 17.1%	60, 1, 64, 24, 24 34.7%, 0.6%, 37.0%, 13.9%, 13.9%	0.881
Treatment planning system HIPLAN/Xio-N	686/231 74.8%/25.2%	570/184 75.6%/24.4%	116/47 71.2%/28.8%	0.360

Abbreviation: PSA: prostate-specific antigen, GS: Gleason score, ADT: androgen-deprivation therapy, Gy: Gray.

**Table 2 cancers-14-04015-t002:** Adverse events after carbon-ion radiotherapy in patients with high-risk prostate cancer.

	Grade 1			Grade 2			Grade 3		
	Overall Study Population	Young Group	Elderly Group	Overall Study Population	Young Group	Elderly Group	Overall Study Population	Young Group	Elderly Group
Radiation dermatitis	17 1.8%	15 2.0%	2 1.2%	1 0.1%	0 0%	1 0.6%	0	0	0
Rectal bleeding	110 11.8%	93 12.3%	17 9.8%	15 1.6%	14 1.9%	1 0.6%	0	0	0
Genitourinary events (Except hematuria)	503 54.1%	399 52.9%	104 60.1%	80 8.6%	65 8.6%	15 8.7%	12 1.3%	11 1.5%	1 0.6%
Hematuria (Number of patients with bladder cancer)	4 0.4%	4 0.5%	0 0%	45 4.8% (6 0.6%)	38 5.0% (5 0.7%)	7 4.0% (1 0.5%)	12 1.3%	11 1.5%	1 0.6%

## Data Availability

The data are not publicly available due to the facility’s privacy policy.

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
