# Peer review of "Safety and Efficacy of Carbon-Ion Radiotherapy for Elderly Patients with High-Risk Prostate Cancer"

_cancers, 2022, doi:10.3390/cancers14164015_

Round 1

Reviewer 1 Report

The manuscript entitled "Safety and efficacy of carbon-ion radiotherapy for elderly patients with high-risk prostate cancer" by Hiroshima et al investigates Carbon ion therapy for high-risk prostate cancer patients focusing on elderly (over 75y) patients. 

The manuscript clearly shows the advantage that Japanese Carbon ion centers have in comparison to centers in the rest of the world. The study comprises a total of 927 patients that were treated over the time period of 18 years. It will be very challenging to compete with such a data set and thus I can highly recommend to publish this study. 

Some minor comments, as listed below, will improve the manuscript and should be considered by the authors: 

Introduction (P2/L66): The part on particle beam therapy is very general and too basic and does not meet the quality of the remaining part of the manuscript.

P2/L71: It remained unclear for me if the Gleason score itself was an inclusion criteria or not (i.e. which value).

P2/L72: "... The efficacy of particle beam therapy has also been reported for the treatment of pancreatic, liver and other cancers". Here I would argue that most of all paediatric cancer patients and H&N patients benefit from particle therapy. 

P2/L76: "... a previous study has reported that CIRT can facilitate further reduction ... " I would say there is more than this one study showing this. 

P2 / L86: "...patients who first underwent CIRT ...". It is not clear if the authors mean patients who got CIRT for the first time or got CIRT before(!) the got another treatment. Please rephrase for clarity. 

Page 3: ADT: According to table 1 all patients got ADT at least for 0.6 months. Please add some more information on the ADT protocols applied and how (if at all) these were adapted between 2000 and 2018.

Page 3 CIRT: The authors could extend this section by some information on the used imaging protocols and how they changed between 2000 and 2018. Also add for clarification what fractionation scheme was applied at which year. As Table 1 provides information on how many patients got which fx scheme, a reference to table 1 should be made in the CIRT chapter. Further add to Table 1 how many patients were planned with XiO-N and how many with HIPLAN. 

Page 4  Line 152: Was the follow-up duration defined on patient individual basis? Please extend the paragraph on follow-up with some more details on the study protocol regarding follow-up. 

Figure 2: The image quality of Figure 2 is very bad. Please improve it for a better readability. 

Page 6 Line 190: Looking at Figure 3 the survival time was also better for those aged 83 , not only 75-79. 

Page 6 Line 185: Typo (omit "were")

Page 7 Line 219: Typo "wel"

Reviewer 2 Report

Thank you for the opportunity to review the manuscript entitled "Safety and efficacy of carbon-ion radiotherapy for elderly patients with high-risk prostate cancer." The authors present an analysis of 927 patients who underwent carbon-ion radiotherapy for high-risk prostate cancer between April 2000 and May 2018. One hundred seventy-three were aged 75 years and older and constituted the elderly group. The authors found the 10-year overall survival, disease-specific survival, biochemical control rate, and biochemical relapse-free survival rates in the young and elderly groups to be 86.9%/71.5%, 96.6%/96.8%, 76.8%/88.1%, and 68.6%/64.3%, respectively. They found that only overall survival (better in the younger group) and biochemical control rate (better in the elderly) met statistical significance. They also found that adverse effects did not differ between the groups. The authors concluded that carbon-ion radiotherapy is safe for elderly and younger patients to treat high-risk prostate cancer. The clinical topic is essential, as optimal treatment can improve survival and morbidity in prostate cancer, the most common cancer in men. However, I have several comments to improve the quality of the manuscript.

  1. Could you expand on the simple summary? It is on the shorter side. You could include methods, quantitative results, conclusions, and possible limitations.
  2. No information in the abstract backs up the conclusion that carbon-ion radiotherapy is safe for young and old patients.
  3. The introduction could be significantly improved. I would allow more space to frame the objective and discuss prior literature, their findings, pros, and cons.
  4. Is there any missing data? If so, what did the authors do about it? 
  5. It is unclear to me what constitutes the young group. Is that everyone who is not 75 or older?
  6. It would benefit the paper if the authors conducted a regression analysis, e.g., by fitting Cox proportional hazard models to the data and identifying men who had no definitive therapy and comparing them against patients who had radiation therapy – and not just comparing 10-year rates (survival rates, biochemical control rate). Then the authors could conduct multivariate analyses controlling for age, Gleason grade group, T stage, year of diagnosis, and/or other relevant variables. 
  7. Line 217: The authors state that because there was no difference in the frequency of adverse events, carbon-ion radiotherapy is safe for both elderly and young patients. I am not sure this makes sense. Just because there is no observed difference between the two groups does not suggest it is safe. Moreover, regarding line 34, the treatment would need to be tested in more diverse cohorts before claiming that this therapy is safe for patients. And what is the definition of safe? Anything that is not CTCAE grade 4 or higher? And if yes, why is that?
  8. Do you have literature backing up the definition of biochemical recurrence in lines 120-121?
  9. I would carefully interpret a high p-value / non-statistically significant findings, meaning there is no difference between groups since the elderly group is relatively small (173 patients). 
  10. In lines 208 and 213, the authors use the word incidence. This typically refers to new cases in a given population. Was it possible that another word was meant to be used? 

Additional minor comments:

- In lines 29 and 30, are the number in parentheses interquartile ranges or ranges?

- Could the authors define high-risk prostate cancer in the methods section? 

- Certain sentences are difficult to interpret and would benefit from proofreading. These include lines 43-46 and line 222. 

- For figure 1, I would change 'whole pt.' with 'all patients. Moreover, it is easier to understand a figure without abbreviations if space permits. 

Round 2

Reviewer 2 Report

Thank you for the updated manuscript.

·     1    It would benefit the paper if the authors rewrite line 23 as it has some grammatical errors

·      2   I am unsure the results of the study support the conclusion in lines 27 and 42 regarding young patients.

·     3    It would benefit the paper if the authors defined young patients in the abstract
